# AN EXPERIMENTAL DESIGN PERSPECTIVE ON MODEL-BASED REINFORCEMENT LEARNING

**Viraj Mehta, Biswajit Paria, & Jeff Schneider**
Robotics Insitute & Machine Learning Department
Carnegie Mellon University
Pittsburgh, PA, USA
{virajm, bparia, schneide}@cs.cmu.edu

**Stefano Ermon & Willie Neiswanger**
Computer Science Department
Stanford University
Stanford, CA, USA
{ermon, neiswanger}@cs.stanford.edu

## ABSTRACT

In many practical applications of RL, it is expensive to observe state transitions from the environment. For example, in the problem of plasma control for nuclear fusion, computing the next state for a given state-action pair requires querying an expensive transition function which can lead to many hours of computer simulation or dollars of scientific research. Such expensive data collection prohibits application of standard RL algorithms which usually require a large number of observations to learn. In this work, we address the problem of efficiently learning a policy while making a minimal number of state-action queries to the transition function. In particular, we leverage ideas from Bayesian optimal experimental design to guide the selection of state-action queries for efficient learning. We propose an *acquisition function* that quantifies how much information a state-action pair would provide about the optimal solution to a Markov decision process. At each iteration, our algorithm maximizes this acquisition function, to choose the most informative state-action pair to be queried, thus yielding a data-efficient RL approach. We experiment with a variety of simulated continuous control problems and show that our approach learns an optimal policy with up to $5 - 1,000\times$ less data than model-based RL baselines and $10^3 - 10^5\times$ less data than model-free RL baselines. We also provide several ablated comparisons which point to substantial improvements arising from the principled method of obtaining data.

## 1 INTRODUCTION

Reinforcement learning (RL) has suffered for years from a curse of poor sample complexity. State-of-the-art model-free reinforcement learning algorithms routinely take tens of thousands of sampled transitions to solve very simple tasks and millions to solve moderately complex ones (Haarnoja et al., 2018; Lillicrap et al., 2015). The current best model-based reinforcement learning (MBRL) algorithms are better, requiring thousands of samples for simple problems and hundreds of thousands of samples for harder ones (Chua et al., 2018). In settings where each sample is expensive, even this smaller cost can be prohibitive for the practical application of RL. For example, in the physical sciences, many simulators require the solution of computationally demanding spatial PDEs in plasma control (Breslau et al., 2018; Char et al., 2019) or aerodynamics applications (Jameson & Fatica, 2006). In robotics, due to the cost of simulating more complicated objects (Heiden et al., 2021), RL methods are typically constrained to fast but limited rigid-body simulators (Todorov et al., 2012). These costly transition functions prompt the question: *"If we were to collect one additional datapoint from anywhere in the state-action space to best improve our solution to the task, which one would it be?"* An answer to this question can be used to guide data collection in RL.

Across the fields of black-box optimization and experimental design, techniques have been developed which choose data to collect that are particularly useful in improving the value of the objective function of the problem. For example, Bayesian optimization (BO) focuses on maximizing an unknown (black-box) function where queries are expensive (Frazier, 2018; Shahriari et al., 2015). More generally, Bayesian optimal experimental design (BOED) aims to choose data to collect which are maximally informative about the value of some derived quantity (Chaloner & Verdinelli, 1995). We aim to leverage these ideas for data-efficiency in reinforcement learning. Along these lines, several

works in the realm of Bayesian RL address this problem in the sequential setting. A Bayes-adaptive MDP (Ross et al., 2007) constructs a modified MDP by augmenting the state space with the posterior belief of the MDP, leading to a policy that can optimally trade off between acquiring more information and exploiting the knowledge it already has. However such an MDP is intractable to exactly solve in large spaces so approximations and heuristics have been developed for the solution (Smith, 2007; Guez et al., 2012). A particularly relevant heuristic is the value of perfect information (VPI) from Dearden et al. (1998), which attempts to capture the potential change in value of a state if the value of a particular action at that state was perfectly known, specifically in a tabular setting. However, VPI doesn't attempt to distinguish between states that are visited during the execution of the optimal policy and those that aren't. This is a critical distinction when collecting data in continuous spaces, as queries may be wasted learning an optimal policy in irrelevant parts of the state space.

Motivated by our opening question, in this paper we study the setting where the agent collects data by sequentially making queries to the transition function with free choice of both the initial state and the action. We refer to this setting as *transition-query reinforcement learning* (TQRL) and formally define it in Section 3.1. Although this setting has been studied in the tabular case, to the best of our knowledge it has not been studied in the continuous MDP literature.

In this work, we draw a connection between MBRL and the world of BOED by deriving an *acquisition function* that quantifies how much information a state-action pair would provide about the optimal solution to a MDP. Like the techniques in Bayesian RL, our acquisition function is able to determine which state-action pairs are worth acquiring in a way which takes into account the reward function and the uncertainty in the dynamics. However, like the Bayes-Adaptive MDP and unlike the VPI heuristic, this function takes into account the current estimates of which states the optimal policy will visit and values potential queries accordingly. Furthermore, our acquisition function is scalable enough to apply to multidimensional continuous control problems. In particular, our acquisition function is the expected information gain (EIG) about the trajectory taken by an optimal policy in the MDP that would be achieved if we were to query the transition function at a given state-action pair.

Finally, we assess the performance of our acquisition function as a data selection strategy in the TQRL setting. Using this method we are able to solve several continuous reinforcement learning tasks (including a nuclear fusion example) using orders of magnitude less data than a variety of competitor methods. In summary, the contributions of our paper are:

- We construct a novel acquisition function that quantifies how much information a state-action pair would provide about the optimal solution to a continuous MDP if the next state were observed from the ground truth transition function. Our acquisition function is able to select relevant datapoints for control purposes leading to improved data efficiency.

- We propose a practical algorithm for computing this acquisition function and use it to solve continuous MDPs in the TQRL setting.

- We evaluate the algorithm on five diverse control tasks, where it is often orders of magnitude more sample-efficient than competitor methods and reaches similar asymptotic performance.

## 2  RELATED WORK

**Transition Query Reinforcement Learning**

In many RL algorithms, data is collected by initializing a policy at a start state and executing actions in the environment in an episodic manner. Kearns et al. (2002) introduced the setting where the agent collects data by sequentially sampling transitions from the ground truth transition model by querying at a state and action of its choice, which they refer to as *RL with access to a generative model*. We refer to this setting for brevity as TQRL. This setting is relevant in a variety of real-world applications where there is a simulator of the transition model available. In particular, we see the setting in nuclear fusion research, where plasma dynamics are modeled by solving large partial differential equations where 200ms of plasma time can take up to an hour in simulation (Breslau et al., 2018).

There is substantial theoretical work on TQRL for finite MDPs. In particular, Azar et al. (2013) give matching log-linear upper and lower PAC sample complexity bounds, a substantial speedup to the upper bound for the standard problem which is quadratic in state size (Kakade, 2003). This is achieved simply by the naive algorithm of learning a transition model by uniformly sampling the space and then performing value iteration on the estimate of the MDP for an optimal policy. More

recently, the bound for this setting was tightened to hold for smaller numbers of samples by Li et al. (2020), meaning that for any dataset size in a continuous problem, the PAC performance can be quantified. Finally, Agarwal et al. (2020) show that the naive 'plug-in' estimator used in the previous works is minimax optimal for this setting. In summary, this setting is thoroughly understood for finite MDPs and it gives a sample complexity reduction from quadratic to linear in the state space size.

To our knowledge there do not exist works specifically solving the TQRL setting for continuous MDPs. In this work, we give an algorithm specifically designed for this setting, which shows sample complexity benefits reminiscent of those theoretically shown in the tabular setting.

**Exploration in Reinforcement Learning** To encourage exploration in RL, agents often use an $\epsilon$-greedy approach (Mnih et al., 2013), upper confidence bounds (UCB) (Chen et al., 2017), Thompson sampling (TS) (Osband et al., 2016), added Ornstein-Uhlenbeck action noise (Lillicrap et al., 2015), or entropy bonuses (Haarnoja et al., 2018) to add noise to a policy which is otherwise optimizing the RL objective. Although UCB, TS, and entropy bonuses all try to adapt the exploration strategy to the problem, they all tackle which action to take from a predetermined state and don't explicitly consider which states would be best to acquire data from.

An ideal method of exploration would be to solve the intractable Bayes-adaptive MDP (Ross et al., 2007), giving an optimal tradeoff between exploration and exploitation. Kolter & Ng (2009); Guez et al. (2012) show that even approximating these techniques in the sequential setting can result in substantial theoretical reductions in sample complexity compared to frequentist PAC-MDP bounds as in Kakade (2003). Other methods stemming from Dearden et al. (1998; 1999) address this by using the myopic value of perfect information as a heuristic for similar Bayesian exploration. However, these methods don't scale to continuous problems and don't provide a way to choose states to query. These methods were further extended with the development of knowledge gradient policies (Ryzhov et al., 2019; Ryzhov & Powell, 2011), which approximate the value function of the Bayes-adaptive MDP, and information-directed sampling (IDS) (Russo & Van Roy, 2014), which takes actions based on minimizing the ratio between squared regret and information gain over dynamics. This was extended to continuous-state finite-action settings in Nikolov et al. (2019). However, this work doesn't solve fully continuous problems, operates in the rollout setting rather than TQRL, and computes the information gain with respect to the dynamics rather than some notion of the optimal policy. In a similar spirit, Arumugam & Van Roy (2021) provide a further generalization of IDS which can also be applied to RL. One recent work very close to ours is Lindner et al. (2021), which actively queries an expensive reward function (instead of dynamics as in this work) to learn a Bayesian model of reward. Another very relevant recent paper (Ball et al., 2020) gives an acquisition strategy in policy space that iteratively trains a data-collection policy in the model that trades off exploration against exploitation using methods from active learning. Achterhold & Stueckler (2021) use techniques from BOED to efficiently calibrate a Neural Process representation of a distribution of dynamics to a particular instance, but this calibration doesn't include information about the task. A tutorial on Bayesian RL methods can be found in Ghavamzadeh et al. (2016) for further reference.

Separate from the techniques used in RL for a particular task, several methods tackle the problem of *unsupervised exploration* (Schmidhuber, 1991), where the goal is to learn as much as possible about the transition model without a task or reward function. One approach synthesizes a reward from modeling errors (Pathak et al., 2017). Another estimates learning progress by estimating model accuracy (Lopes et al., 2012). Others use an information gain-motivated formulation of model disagreement (Pathak et al., 2019; Shyam et al., 2019) as a reward. Other methods incentivize the policy to explore regions it hasn't been before using hash-based counts (Tang et al., 2017), predictions mimicking a randomly initialized network (Burda et al., 2019), a density estimate (Bellemare et al., 2016), or predictive entropy (Buisson-Fenet et al., 2020). However, these methods all assume that there is no reward function and are inefficient for the setting of this paper, as they spend time exploring areas of state space which can be quickly determined to be bad for maximizing reward on a task.

**Bayesian Algorithm Execution and BOED**

Recently, a flexible framework known as Bayesian algorithm execution (BAX) (Neiswanger et al., 2021) has been proposed for efficiently estimating properties of expensive black-box functions, which builds off of a large literature from Bayesian Optimal Experiment Design (Chaloner & Verdinelli, 1995). The BAX framework gives a general procedure for sampling points which are informative about the future execution of an algorithm. In this paper, we extend this framework to the setting

of model-predictive control, when we have expensive dynamics (i.e. transition function) which we treat as a black-box function in the BAX framework. Via this strategy, we are able to use similar techniques to develop acquisition functions for data collection in reinforcement learning.

**Gaussian Processes (GPs) in Reinforcement Learning** There has been substantial prior work using GPs in reinforcement learning. Most well-known is PILCO (Deisenroth & Rasmussen, 2011), which computes approximate analytic gradients of policy parameters through the GP dynamics model while accounting for uncertainty. Most related to our eventual MPC method is (Kamthe & Deisenroth, 2018), which gives a principled probabilistic model-predictive control algorithm for GPs.

## 3 PRELIMINARIES

In this work we deal with finite-horizon discrete-time *Markov decision processes* (MDPs) which consist of a tuple $\langle \mathcal{S}, \mathcal{A}, T, r, p_0, H \rangle$ where $\mathcal{S}$ is the state space, $\mathcal{A}$ is the action space, $T$ is the transition function $T : \mathcal{S} \times \mathcal{A} \to P(\mathcal{S})$ (using the convention that $P(\mathcal{X})$ is the set of probability measures over $\mathcal{X}$), $r : \mathcal{S} \times \mathcal{A} \times \mathcal{S} \to \mathbb{R}$ is a reward function, $p_0(s)$ is a distribution over $\mathcal{S}$ of start states, and $H \in \mathbb{N}$ is a horizon. We always assume $\mathcal{S}, \mathcal{A}, p_0, H$ are known. We also assume the reward $r$ is known, though our development of the method can easily be generalized to the case where $r$ is unknown. Our primary function of interest is the transition function $T$, which we learn from data. Our aim is to find a policy $\pi : \mathcal{S} \to P(\mathcal{A})$ that maximizes the objective given below. We can describe the execution of $\pi$ in the MDP as a finite collection of random variables generated by $s_0 \sim p_0$ and $a_i \sim \pi(s_i), s_i \sim T(s_{i-i}, a_{i-1})$ for $i \in 1, \ldots, H$. Then this objective can be written

$$J_T(\pi) = \mathbb{E}_{p(s_{0:H}, a_{0:H-1})} \left[ \sum_{i=0}^{H-1} r(s_i, a_i, s_{i+1}) \right]. \tag{1}$$

We aim to maximize this objective while minimizing the number of samples from the ground truth transition function $T$ that are required to reach good performance. We denote the optimal policy as $\pi^* = \text{argmax}_\pi J_T(\pi)$, which we can assume to be deterministic (Sutton & Barto, 1998) but not necessarily unique. Finally, we assume that there is some prior $P(T)$ for which the posterior $P(T \mid D)$ is available for sampling given a dataset $D$.

### 3.1 TRANSITION QUERY REINFORCEMENT LEARNING (TQRL)

In the standard online RL setting, one assumes data $D = \{(s_i, a_i, s'_i)\}_{i \in [n]}$ must be collected in length-$H$ trajectories (*rollouts*) where the initial state $s_0 \sim p_0$, and after an action $a_i$ is chosen, the next state $s'_i = s_{i+1}$ is sampled from $T(s_i, a_i)$ up to $i = H$, at which point the process repeats.

In this work, we consider the TQRL setting, where the agent sequentially acquires data $(s_i, a_i, s'_i)$ in arbitrary order by querying a state action pair $(s_i, a_i)$ from $\mathcal{S} \times \mathcal{A}$ and recieving a sample $s'_i \sim T(s, a)$ from the black-box transition function $T$ (Kearns et al., 2002; Kakade, 2003; Azar et al., 2013). The goal in both settings is to find a policy which optimizes the objective in Equation (1).

It has been shown for finite MDPs in (Azar et al., 2013) that the PAC sample complexity, which is the number of samples required to identify with high probability a policy that achieves almost optimal value, of this setting is $\tilde{O}(|\mathcal{S}||\mathcal{A}|)$, ignoring the PAC factors. This is notably better than the bound of $\tilde{O}(|\mathcal{S}|^2|\mathcal{A}|)$ in the online RL setting given in Section 8.3 of Kakade (2003). The improvement shown in finite cases suggests that there could be similar reductions available in a continuous setting.

### 3.2 MODEL-PREDICTIVE CONTROL

Though our acquisition function is derived for any policy search procedure including planning or model-free reinforcement learning algorithms, we focus on model-predictive control (MPC) as it is simple and requires minimal components to function. MPC is a standard technique in model-based reinforcement learning (Chua et al., 2018; Wang & Ba, 2020). Using an estimated dynamics model $\hat{T} : \mathcal{S} \times \mathcal{A} \to \mathcal{S}$ and an optimization algorithm, an MPC strategy with a planning horizon $h \in \mathbb{N}$ choosing an action at a state $s_0$ solves the planning problem

$$\max_{a_0, \ldots, a_h \in \mathcal{A}} \mathbb{E}_{s_{i+1} \sim \hat{T}(s_i, a_i)} \left[ \sum_{i=0}^{h} r(s_i, a_i, s_{i+1}) \right]. \tag{2}$$

Planning is typically redone periodically, often every timestep, as actions are executed in the real environment. The cross-entropy method (CEM) is often used to solve this optimization problem. In this work, we use an improved variant of CEM described in Pinneri et al. (2020). Here we will refer to $\pi_T$ as the stochastic policy obtained by running MPC over a dynamics function $T$. The randomness in the policy is due to any randomness in $T$ and the randomness used by the optimizer. We give details on the hyperparameters involved in this optimization problem and their values in Section B.

## 4 AN ACQUISITION FUNCTION FOR MODEL-BASED RL

We draw inspiration from BO and BOED in constructing an acquisition function suitable for control applications. For our purposes, an *acquisition function* is a computationally tractable function $\mathcal{S} \times \mathcal{A} \to \mathbb{R}$ that describes the marginal improvement in the performance of the policy on the MDP (conditioned on all previously observed data) when observing one additional state-action pair $(s, a) \in \mathcal{S} \times \mathcal{A}$. As acquisition functions are greedy, they aren't necessarily optimal data-selection strategies given a fixed budget, compared to non-myopic strategies such as solving the Bayes-adaptive MDP. However, greedy strategies using mutual information are tractable and are often effective due to the submodularity of the expected information gain. More specifically, our acquisition function is an expected information gain (EIG) or equivalently the mutual information (MI) between a query of our transition model and a representation of the optimal policy, as we elaborate in this section.

A typical approach in a Bayesian setting might be to gather data such that the entropy $\mathbb{H}[\pi^*]$ of the belief of the optimal policy $\pi^*$ is minimized. However, a full specification of $\pi^*$ includes the behavior of the policy in all parts of the state space including states that are not visited at all, or visited less often in rollouts of $\pi^*$ when the start state is sampled from $p_0$. As a result, not all points in the state space are equally important when learning an optimal policy aimed at maximizing the expected reward. Optimizing the entropy of $\pi^*$ would lead to a uniform treatment of all the points in the state space, and hence would be far from optimal for the standard goal in RL.

We instead propose to minimize the entropy of the *optimal trajectory* $\tau^* = \{s_i\}_{i=1}^H$ defined as a random vector of states generated by first sampling $s_0 \sim p_0$ then sampling $a_i = \pi^*(s_i), s_{i+1} \sim T(s_i, a_i)$ for $H$ timesteps. The optimal trajectory is completely specified by $\pi^*$ and the randomness arising from the MDP. Furthermore, $\tau^*$ contains the necessary information needed about the transition function $T$ to solve the MDP, since any state that could ever be visited by $\pi^*$ is in the support of $\tau^*$. We empirically observe that this leads to an efficient strategy for active RL.

The randomness in $\tau^*$ arises from three sources: the start state distribution $p_0$, the dynamics $T$ constituting the *aleatoric uncertainty*, and the uncertainty in our estimate of the model $T$ due to our limited experience which constitutes the *epistemic uncertainty*. The first two sources of uncertainty being aleatoric in nature cannot be reduced by experience. Our proposed acquisition function based on information gain naturally leads to reduction in the epistemic uncertainty about $\tau^*$ as desired. Finally, our acquisition function for a given state-action pair $(s, a)$ is given as

$$
\begin{aligned}
\text{EIG}_{\tau^*}(s, a) &= \mathbb{E}_{s' \sim T(s,a|D)} \Big[ \mathbb{H}[\tau^* \mid D] - \mathbb{H}[\tau^* \mid D \cup \{(s, a, s')\}] \Big] \\
&= \mathbb{E}_{s_0 \sim p_0} \Big[ \mathbb{E}_{s' \sim T(s,a|D)} \Big[ \mathbb{H}[\tau^* \mid D, s_0] - \mathbb{H}[\tau^* \mid D \cup \{(s, a, s')\}, s_0] \Big] \Big].
\end{aligned}
\tag{3}
$$

Here we assume a posterior model of the dynamics $T(s, a \mid D)$ for a dataset $D$ we have observed. The second equality is true because $s_0 \perp s' \mid s, a$. In this paper, we assume the MPC policy using the ground truth transition function is approximately optimal, i.e. $\pi_T \approx \pi^*$, though in principle $\pi^*$ could be approximated using any method. Of course, our method never actually has access to $\pi_T$ or $\pi^*$.

### 4.1 ESTIMATING $\text{EIG}_{\tau^*}$ VIA POSTERIOR FUNCTION SAMPLING

For $\text{EIG}_{\tau^*}$ to be of practical benefit, we must be able to tractably approximate it. Here we show how to obtain such an approximation. By the symmetry of MI, we can rewrite Equation (3) as

$$
\text{EIG}_{\tau^*}(s, a) = \mathbb{E}_{s_0 \sim p_0} \left[ \mathbb{E}_{\tau^* \sim P(\tau^*|D)} [\mathbb{H}[s' \mid s, a, D, s_0] - \mathbb{H}[s' \mid s, a, \tau^*, D, s_0]] \right].
\tag{4}
$$

Since $\mathbb{H}[s' \mid s, a, D, s_0] = \mathbb{H}[s' \mid s, a, D]$ doesn't depend on $\tau^*$ or $s_0$, we can simply compute it as the entropy of the posterior predictive distribution $P(s' \mid s, a, D)$ given by our posterior over the transition function $P(T \mid D)$. In order to compute the other term, we must take samples $\tau_{ij}^* \sim P(\tau^* \mid D)$. To do this, we first sample $m$ start states $s_0^i$ from $p_0$ (we always set $m = 1$ in

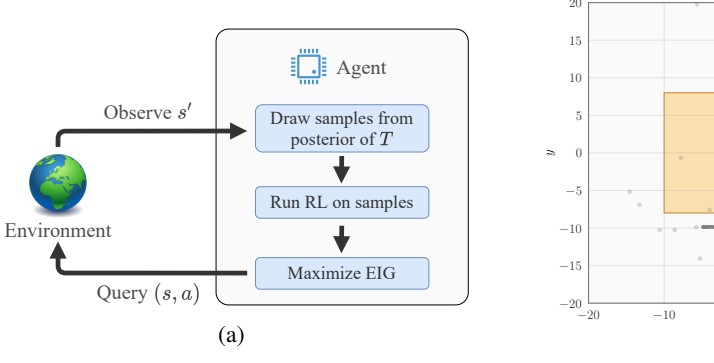

(a)

(b)

Figure 1: (a) A diagram of the BARL data-collection loop. (b) An illustration of the $\text{EIG}_{\tau^*}$ computation over several sample paths $\tau_i^*$ (multi-colors) sampled from $P(\tau^* \mid D)$ for a dataset of past queries (grey points). The optimizer (in pink) is a point that is maximally informative when learning a model for crossing the path between the lava pools (orange rectangles) to the goal (green).

experiments but derive the procedure in general) and for each start state independently sample $n$ posterior functions $T'_{ij} \sim P(T' \mid D)$ from our posterior over dynamics models. We then run the MPC procedure on each of the posterior functions from $s_0^i$ using $T'_{ij}$ for $T$ and $\pi_{T'_{ij}}$ for $\pi^*$ (using our assumption that $\pi^* \approx \pi_T$), giving our sampled $\tau_{ij}^*$. This is an expression of the generative process for $\tau^*$ as described in the previous section that accounts for the uncertainty in $T$. Formally, we can approximate $\text{EIG}_{\tau^*}$ via Monte-Carlo as

$$\text{EIG}_{\tau^*}(s, a) \approx \mathbb{H}[s' \mid s, a, D] - \frac{1}{mn} \sum_{i \in [m]} \sum_{j \in [n]} \mathbb{H}[s'|s, a, \tau_{ij}^*, D]. \tag{5}$$

Finally, we must calculate the entropy $\mathbb{H}[s'|s, a, \tau_i^*, D]$. For this, we follow a similar strategy as Neiswanger et al. (2021). In particular, since $\tau_i^*$ is a set of states output from the transition model, we can treat them as additional noiseless datapoints for our dynamics model and condition on them. In the following section we describe our instantiation of this EIG estimate, and how we can use it in reinforcement learning procedures. Though inspired by the work cited here, we modify the computation of the acquisition function to factor $p_0$ as an irreducible source of uncertainty. We also extend the function being queried to be vector-valued.

## 5  BAYESIAN ACTIVE REINFORCEMENT LEARNING

In this work, we take a simple approach for nonlinear control in continuous spaces and assume a Gaussian process (GP) prior $P(T)$ to model the dynamics. Though computationally expensive, this choice ensures that we can easily approximate all necessary quantities. However, we note that the development of the acquisition function is general and any Bayesian model could be used in principle.

The transition function $T : \mathcal{S} \times \mathcal{A} \to p(\mathcal{S})$ (dynamics) can be modeled with a GP due to its non-parametric nature and ability to capture uncertainties in $T$. The transition function takes a state action pair $(s, a) \in \mathbb{R}^{d+n}$ as input, and produces a $d$-dimensional output denoting the next state. We model each of the $d$ dimensions of the output as independent GPs. More specifically, we model the change in state $\Delta(s, a) = T(s, a) - s$ rather than the final state $T(s, a)$ directly. This is helpful for continuous control problems since the state often changes by only a small magnitude. Given observations $D = \{(s_i, a_i, s'_i)\}$, our approach requires a posterior sample of the transition function conditioned on $D$. We follow the approach of Wilson et al. (2020), based on sparse-GPs and random fourier approximations of kernels (Rahimi et al., 2007), allowing us to approximately but efficiently sample from the GP posterior conditioned on the observations.

Assuming access to a generative model and an initial dataset $D$ (for which, in practice, we use one randomly sampled datapoint $(s, a, s')$), we compute $\text{EIG}_{\tau^*}$ for $D$ by running MPC on posterior function samples and approximate $\text{argmax}_{s \in \mathcal{S}, a \in \mathcal{A}} \text{EIG}_{\tau^*}(s, a)$ by zeroth order approximation. Then we query $s' \sim T(s, a)$ and add the subsequent triple to the dataset $D$ and repeat the process. To evaluate, we simply perform the MPC procedure in Equation (2) and execute $\pi_{\mathbb{E}[T|D]}$ on the real

environment. We refer to this procedure as Bayesian active reinforcement learning (BARL). Details are given in Algorithm 1 (here, $U$ denotes the uniform distribution) and a schematic diagram in Figure 1a. We discuss details of training hyperparameters and the GP model in Appendix A.

---

**Algorithm 1** Bayesian active reinforcement learning (BARL) using $\mathrm{EIG}_{\tau^*}$

---

**Inputs:** transition function query budget $b$, number of points for optimization $k$, number of posterior function samples $n$.
Initialize $(x_0, y_0) \sim U(\mathcal{S} \times \mathcal{A})$, $x'_0 \sim T(x_0, y_0)$, and $D \leftarrow \{(x_0, y_0, x'_0)\}$.
**for** $i \in [b]$ **do**
    Sample posterior functions $\{T'_\ell\}_{\ell=1}^n \sim P(T' \mid D)$.
    Sample start state $s_0 \sim p_0$.
    Compute $\{\tau^*{}_\ell\}_{\ell=1}^n$ by executing MPC policy $\pi_{T'_\ell}$ on the dynamics of $T'_\ell$ starting from $s_0$.
    $(x_1, y_1), \ldots, (x_k, y_k) \sim U(\mathcal{S} \times \mathcal{A})$.
    $(x^*, y^*) \leftarrow \mathrm{argmax}_{\{(x_i, y_i)\}_{i=1}^k} \mathrm{EIG}_{\tau^*}(x_i, y_i)$.
    $D \leftarrow D \cup \{(x^*, y^*, x'^*)\}$ where $x'^* \sim T(x^*, y^*)$.
**end for**
**return** $\pi_{\hat{T}}$ where $\hat{T}(s, a) = \mathbb{E}_{T \sim P(T|D)}[T(s, a)]$

---

## 6 EXPERIMENTS

The aim of our study of acquisition functions for RL is to reduce the sample complexity of learning good policies in continuous spaces, under expensive dynamics. Here, we demonstrate the effectiveness of using $\mathrm{EIG}_{\tau^*}$ to leverage transition queries by comparing against a variety of state-of-the-art RL algorithms. In particular, we compare the average return across five evaluation episodes across five runs with differing random seeds of each algorithm on five continuous control problems as data is collected. We also assess the amount of data taken by each algorithm to 'solve' the problem, which is taken to mean performing as well as our MPC procedure using the ground truth dynamics. Our proposed method, BARL, greatly outperforms other methods across the board. In particular, BARL uses $5 - 1,000\times$ less data to solve problems than state-of-the-art model-based RL algorithms and $10^3 - 10^5\times$ less data than model-free RL algorithms. In this section we primarily focus on the performance of the controller, and in section A.1 we also discuss the runtime of the algorithm.

**Comparison Methods.** We use as our model-based comparison methods in this work PETS (Chua et al., 2018) as implemented by Pineda et al. (2021), which does MPC using a probabilistic ensemble of neural networks and particle sampling for stochastic dynamics and a similar MPC method using the mean of the same GP model we use for BARL to execute $\pi_{\hat{T}}$ to collect data as in the standard RL setting. We also compare against PILCO (Deisenroth & Rasmussen, 2011), which also leverages a GP to directly optimize a policy that maximizes an uncertainty-aware long term reward. For model-free methods, we use Soft Actor-Critic (SAC) (Haarnoja et al., 2018), which is an actor-critic method that uses an entropy bonus for the policy to encourage evaluation, TD3 (Fujimoto et al., 2018) which addresses the stability questions of actor-critic methods by including twin networks for value and several other modifications, and Proximal Policy Optimization (PPO) (Schulman et al., 2017), which addresses stability by forcing the policy to change slowly in KL so that the critic remains accurate. As a baseline TQRL method and to better understand the GP performance, we use a method we denote $\mathrm{EIG}_T$, which chooses points which maximize the predictive entropy of the transition model to collect data. We believe that when given access to transition queries many unsupervised exploration methods like Pathak et al. (2019); Shyam et al. (2019) or methods which value information gain over the transtion function (Nikolov et al., 2019) would default to this behavior.

**Control Problems.** We tackle five control problems: the standard underactuated **pendulum** swing-up problem (Pendulum-v0 from Brockman et al. (2016)), a **cartpole** swing-up problem, a 2D **lava path** navigation problem, a 2-DOF robot arm **reacher** problem with 8-dimensional state (Reacher-v2 from Brockman et al. (2016)), and a simplified **beta tracking** problem from plasma control (Char et al., 2019; Mehta et al., 2020) where the controller must maintain a fixed normalized plasma pressure using as GT dynamics a model learned similarly to Abbate et al. (2021). The lava path is intended to test stability and exploration of algorithms. The goal is to reach a fixed goal state from a narrow uniform distribution over start states. As shown in Figure 1b, the state space contains a 'lava' region which gives large negative rewards for every timestep. When not in lava, the reward is simply the negative squared distance to the goal, forcing the agent to navigate to the goal as quickly as possible.

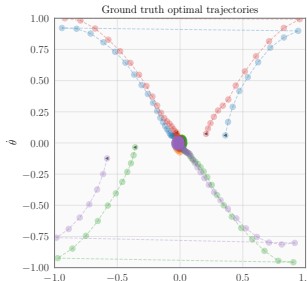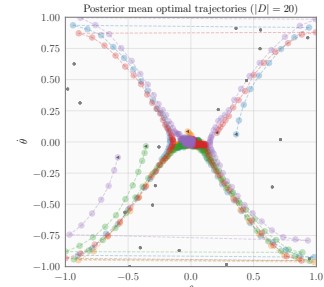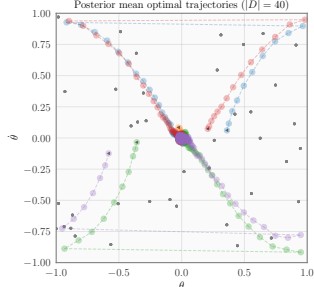

Figure 2: Progress and sampled points of BARL, showing trajectories through the normalized state space of the pendulum problem from four fixed start points for the optimal controller and MPC with 20 and 40 datapoints, respectively. It is clear that even with very few points the controller is able to closely track the optimal paths. Here color is used only to disambiguate the trajectories.

| Environment | BARL | MPC | $\text{EIG}_T$ | PETS | SAC | TD3 | PPO |
|---|---|---|---|---|---|---|---|
| Lava Path | **11** | 41 | 41 | 600 | N/A | N/A | N/A |
| Pendulum | **16** | 46 | 46 | 5200 | 6000 | 57000 | 13000 |
| Cartpole | **91** | 161 | 121 | 1625 | 31000 | 18000 | N/A |
| Beta Tracking | 96 | **36** | N/A | 300 | 9000 | 6000 | 16000 |
| Reacher | **251** | 751 | N/A | 700 | 23000 | 13000 | N/A |

Table 1: **Sample Complexity:** Median number of samples across 5 seeds required to reach 'solved' performance, averaged across 5 trials. We determine 'solved' performance by running an MPC policy (similar to the one used for evaluation) on the ground truth dynamics to predict actions. We record 'N/A' when the median run is unable to solve the problem by the end of training.

Since there is a narrow path through the lava, we want to explore a policy which crosses efficiently and safely. Agents who fail to find this solution will be forced to go around, incurring penalties.

We see in both the sample complexity figures in Table 1, the learning curves in Figure 3, and visually in Figure 2 that BARL leverages $\text{EIG}_{\tau^*}$ to significantly reduce the data requirements of learning controllers on the problems presented. We'd like to additionally point out several failure cases of related algorithms that BARL avoids. Though it performs well on the simplest environments (pendulum and cartpole), $\text{EIG}_T$ suffers from an inability to focus on acquiring data relevant to the control problem and not just learning dynamics as the state space becomes higher-dimensional in the reacher problem, or less smooth as in the beta tracking problem. The MPC method performs reasonably well across the board and is competitive with BARL on the plasma problem but requires relatively more samples in smaller environments where the model uncertainty can point to meaningfully underexplored areas. PETS is strong across the board but suffers from more required samples due to both its neural network dynamics model and its inability to make transition queries.

All algorithms besides BARL suffer substantial instability on the lava path problem, which is designed to be challenging to explore in a sequential fashion and require a precise understanding of which areas are safe to enter. BARL manages to learn where it is safe to operate in a handful of queries, which is an exciting result and will bear further investigation. Figure 1b gives some intuition as to why: points are initally queried close to the start and as those dynamics are understood they are subsequently queried farther and farther along the execution paths. This allows BARL to use transition queries to avoid traversing well-understood areas of state space to reach the areas which are worth learning. We see a speedup in sample complexity reminiscent of a move from quadratic to linear, which mirrors some of the theoretical improvements given in the prior work discussed on tabular methods.

We further support our assertion that BARL is picking 'meaningful' points to the control problem by the evidence in Figure 4. Here, BARL is able to solve the reacher problem while $\text{EIG}_T$ is not. However, BARL has much worse model predictions on random data than $\text{EIG}_T$ while doing a much better job modeling data used by the MPC procedure. Clearly, the $\text{EIG}_{\tau^*}$ acquisition function captures in some way which data would be valuable to acquire to not just learn about the transition function but actually solve the control problem. We see this pattern across other tasks as well. In

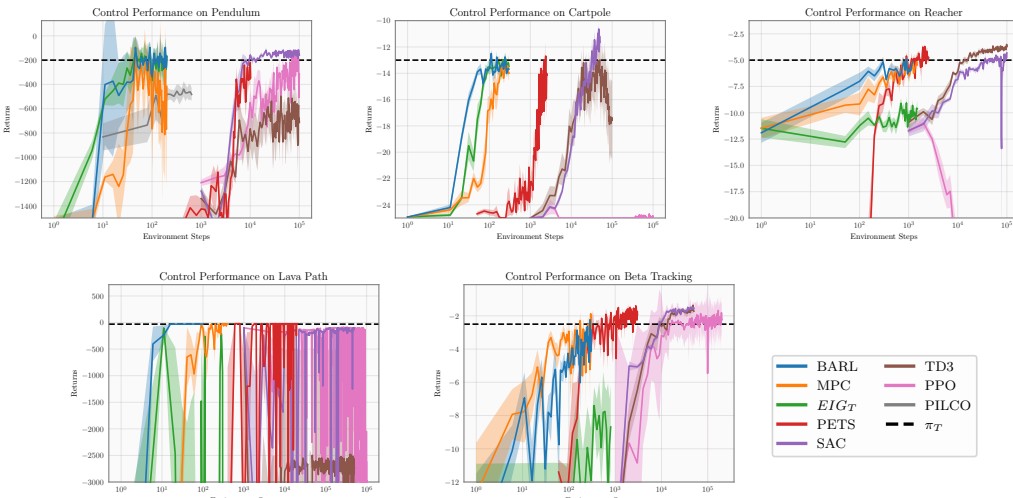

Figure 3: Learning Curves of RL methods, showing control performance averaged across 5 seeds. In each, the $x$-axis is on a logarithmic scale to account for widely varying data requirements. We see that though most algorithms end up reaching roughly the same performance on each task, BARL is substantially more efficient in most cases. The shaded region is the standard error of the average performance across the 5 seeds. We additionally include a plot of the performance of the PILCO algorithm (Deisenroth & Rasmussen, 2011) on Pendulum. PILCO makes assumptions about the initial state distribution and suffers from numerical instability under long control horizon so we were unable to reach representative performance on the other problems.

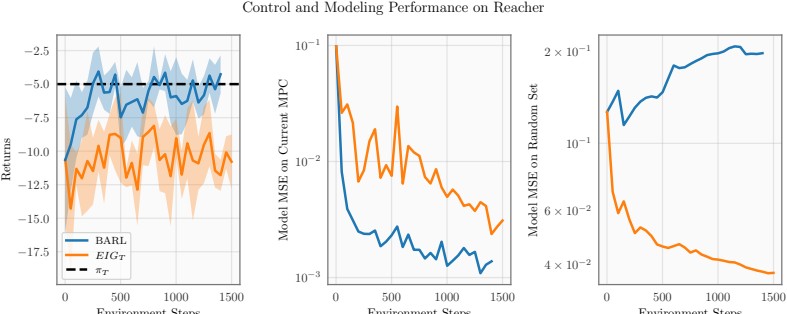

Figure 4: For a single run of BARL and of $\text{EIG}_T$ using the same prior model, we evaluate control performance, as well as modeling error, on both the predictions used by the MPC procedure and on a set of points uniformly sampled from the state-action space.

section B.1, we study whether the acquisition function we see here is able to work with a suboptimal controller on posterior samples of the dynamics. Our experiments show that $\text{EIG}_{\tau^*}$ seems to work well even when the policy used to generate $\tau^*$ is suboptimal.

## 7 DISCUSSION AND FUTURE WORK

In this work, we proposed an acquisition function for reinforcement learning, and applied it to the setting of TQRL, leading to a novel algorithm for addressing the problem of efficient data collection in RL. We experimented with several control problems and demonstrated that this approach leads to substantial improvements with respect to the sample efficiency.

However, there are some drawbacks to this method as well. Computing the proposed acquisition function relies on executing the entire control algorithm over a set of posterior function samples, leading to high computational requirements. In our current implementation we use Gaussian processes to model the transition function, which are computationally expensive and do not scale well to higher dimensions and large datasets. In the future, we plan to extend this idea to use other types of Bayesian models such as Bayesian neural networks and take advantage of GPU compute for better scalability.

ACKNOWLEDGMENTS

We would like to acknowledge anonymous reviewers for valuable feedback. VM acknowledges Swapnil Pande for providing the idea for and implementation of the Lava Path environment and Ian Char for providing the trained model for the Beta Tracking environment. This work was funded in part by DOE grant number DE-SC0021414. WN acknowledges the helpful feedback from members of the Ermon Group studying RL. WN was supported in part by NSF (#1651565), ONR (N000141912145), AFOSR (FA95501910024), ARO (W911NF-21-1-0125), DOE (DE-AC02-76SF00515) and Sloan Fellowship.

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

| Control Problem | Pendulum | Cartpole | Lava Path | Reacher | Beta Tracking |
|---|---|---|---|---|---|
| Budget $b$ | 200 | 300 | 100 | 1500 | 300 |
| # of points for optimization $k$ | 1000 | 1000 | 1000 | 1000 | 1000 |
| # of posterior function samples $n$ | 15 | 15 | 15 | 15 | 15 |

Table 2: BARL hyperparameters used for each control problem.

| Control Problem | Pendulum | Cartpole | Lava Path | Reacher | Beta Tracking |
|---|---|---|---|---|---|
| Sample $\tau^*$ $n$ times | 20.7 - 18.5 | 38.7 - 33.9 | 33.4 - 35.8 | 216 - 276 | 7.5 - 4.9 |
| Evaluate $\text{EIG}_{\tau^*}$ at $k$ points | 3.3 - 5.5 | 7.5 - 12.7 | 7.6 - 9.3 | 21.3 - 147 | 7.5 - 13.6 |
| Total for BARL Iteration | 24 - 24.04 | 46.2 - 46.5 | 40.1 - 45.1 | 237 - 423 | 15 - 18.5 |
| Evaluation for one episode | 7.2 - 21.5 | 2.5 - 10.3 | 18 - 47.9 | 26.2 - 913.7 | 0.9 - 3.5 |

Table 3: Runtime in seconds for the phases of the BARL algorithm on all problems when run on the author's 24-core CPU machines. The ranges given show the runtime for the operation at the beginning and at the end of training, as some operations run longer as more data is added.

## A  TRAINING DETAILS

We vary our budget based on our understanding of how much data would be required to solve the problem. The other hyperparameters of the BARL algorithm are constant but listed for completeness in Table 2.

For all of our experiments, we use a squared exponential kernel with automatic relevance determination (MacKay et al., 1994; Neal, 1995). The parameters of the kernel were estimated by maximizing the likelihood of the parameters after marginalizing over the posterior GP (Williams & Rasmussen, 1996).

To optimize the transition function, we simply sampled a set of points from the domain, evaluated the acquisition function, and chose the maximum of the set. This set was chosed uniformly for every problem but Reacher, for which we chose a random subset of $\cup_i \cup_j \tau_{ij}^*$ (the posterior samples of the optimal trajectory) since the space of samples is 10-dimensional and uniform random sampling will not get good coverage of interesting regions of the state space.

### A.1  RUNTIME DETAILS

Based on these choices and the MPC hyperparameters below in Section B, each of these problems results in a varying runtime for the BARL algorithm. In Table 3, we report the time taken for an iteration of BARL and how it breaks down by step. We give these as ranges, as the computational time requires increases as the learning process continues since GP computational costs scale with the size of the dataset. We also include for completeness the time taken to execute the MPC policy on the ground truth problem, which is not strictly part of the BARL algorithm but still relevant to practitioners.

Clearly, BARL is a relatively slow algorithm computationally. But in settings where samples are scarce, BARL is much cheaper than alternative methods which might use less compute for the RL algorthms but require many more samples. When compared to the costs of running an hour-long simulation or running a costly experiments, spending a few minutes computing the acquisition function seems like a good use of resources.

## B  MPC DETAILS

As we've discussed, we use model-predictive control in this work to choose actions which maximize future reward given a model of the dynamics. In particular we use the improved Cross-Entropy Method from Pinneri et al. (2020) to solve the optimization problem in Equation 2, which uses several tricks including colored noise samples and caching to reduce the number of queries to the planning

| Control Problem | Pendulum | Cartpole | Lava Path | Reacher | Beta Tracking |
|---|---|---|---|---|---|
| Base number of samples | 25 | 30 | 25 | 100 | 25 |
| Number of elites | 3 | 6 | 4 | 15 | 3 |
| Planning horizon | 20 | 15 | 20 | 15 | 5 |
| Number of iCEM iterations | 3 | 5 | 3 | 5 | 3 |
| Replanning Period | 6 | 1 | 6 | 1 | 2 |

Table 4: Hyperparameters used for optimization in MPC procedure for control problems.

model. There is a natural trade-off in any search method between computational cost and quality of actions found in terms of predicted reward. In this work, we chose hyperparameters for each task that were as computationally light as possible which attained a similar reward to larger hyperparameters when executing MPC using the ground-truth model ($\pi_T$, in our terms). As recommended by the original paper, we use $\beta = 3$ for the scaling exponent of the power spectrum density of sampled noise for action sequences, $\gamma = 1.25$ for the exponential decay of population size, and $\xi = 0.3$ for the amount of caching.

We manually tuned the base number of samples, planning horizon, number of elites to take from the sampled action sequences, number of iterations of planning, and the replanning period of the model. Here we give the ultimate values for those parameters, which were used for all ablations using our GP model and MPC. The values we used across all experiments for each problem are given in Table 4.

### B.1 Robustness of $\mathrm{EIG}_{\tau^*}$ to a suboptimal controller

In order to compute $\mathrm{EIG}_{\tau^*}$ in this work, we perform model-predictive control on posterior transition function samples (execute $\pi_{T'_\ell}$ on $T'_\ell$, in our notation). We assume that $\pi_{T'_\ell}$ is close to the optimal policy for the MDP with transition function $T'_\ell$. However, this assumption could lead to pathologies in the method if it doesn't hold in practice. In this section, we emprically investigate the consequences of using a suboptimal controller when finding samples of $\tau^*$ on posterior samples of the transition function.

In order to understand the sensitivity of $\mathrm{EIG}_{\tau^*}$ to the MPC policy executed on posterior samples, we ran experiments where we reduced the planning budget or horizon for the posterior function policy in order to see whether the acquisition function fails. In particular, we ran the reacher and cartpole experiments from the main paper with varying MPC budgets for the posterior function policy $\pi_{T'_\ell}$ and a fixed MPC budget at test time. This allows us to isolate the effect of a suboptimal policy generating samples of $\tau^*$.

On the reacher experiment, we vary the number of CEM iterations ranging from 1 to 5. This is straightforwardly linked to the amount of search the policy conducts before executing an action. On the cartpole experiment, we varied the length of the planning horizon, which affects how far in the future the policy will consider actions as it is deciding what is the good immediate next action. In both cases, we see in Figure 5 that performance hardly changes as the budget for MPC is reduced. Only on the reacher problem when the number of CEM iterations is reduced to 1 (effectively reducing CEM to simple random search) do we see a significant drop in performance. This supports the notion that the quality of the approximation of the optimal policy in $\mathrm{EIG}_O$ is not critical to the performance of the acquisition function as a data selection strategy. We also plot in the figure the performance of an MPC controller on the ground truth dynamics with these reduced MPC budgets. It is clear that if we were to execute these degraded policies at test time, they would be much worse. We find it interesting and intend to further study the robustness of using cheaper policies to decide where to acquire data.

## C Description of Continuous Control Problems

**Lava Path.** The lava path has 4-dimensional state (position and velocity) and 2-dimensional action (an applied force in the plane). The goal is to reach a fixed goal state from a relatively narrow uniform distribution over start states. As shown in Figure 1b, the state space contains a 'lava' region which

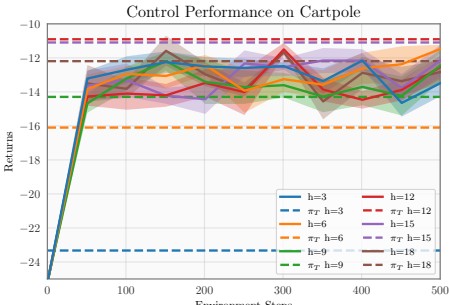 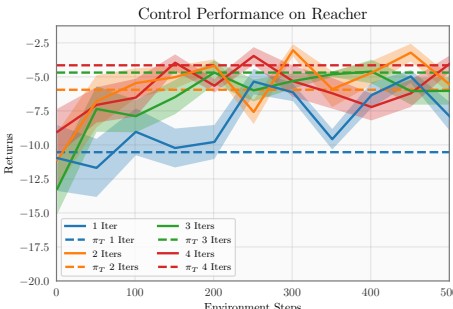

Figure 5: Performance of BARL when MPC budget for posterior function samples is varied while MPC test time budget is held constant. The error regions are the standard error of the return seen across 5 trials of the policy. The dashed lines are the performances that MPC with the equivalent hyperparameters achieves if executed at test time given the ground truth dynamics.

gives very large negative rewards for every timestep. Other than when in lava, the reward is simply the negative squared distance to the goal, forcing the agent to navigate to the goal as quickly as possible. Since the lava has a narrow path through, the actor is forced to explore a policy which will realize that it is safe and efficient to cross. Agents who fail to find this solution will be forced to go around, incurring penalties.

**Pendulum.** The pendulum swing-up problem is the standard one found in the OpenAI gym (Brockman et al., 2016). The state space contains the angle of the pendulum and its first derivative and action space simply the scalar torque applied by the motor on the pendulum. The challenge in this problem is that the motor doesn't have enough torque to simply rotate the pendulum up from all positions and often requires a back-and-forth swing to achieve a vertically balanced position. The reward function here penalizes deviation from an upright pole and squared torque.

**Cartpole.** The cartpole swing-up problem has 4-dimensional state (position of the cart and its velocity, angle of the pole and its angular velocity) and a 1-dimensional action (horizontal force applied to the cart). Here, the difficulty lies in translating the horizontal motion of the cart into effective torque on the pole. The reward function here is a negative sigmoid function penalizing the distance between the tip of the pole and a centered upright goal position.

**Reacher.** The reacher problem simulates a 2-DOF robot arm aiming to move the end effector to a randomly resampled target provided. The problem requires joint angles and velocities as well as an indication of the direction of the goal, giving an 8-dimensional state space with the mentioned 2-D control. Our results on this problem are particularly encouraging as they show that BARL can scale to some problems with higher dimensionalities.

**Beta Tracking (Nuclear Fusion).** Finally, the beta tracking problem has 4-dimensional state consisting of the current normalized plasma performance $\beta_N$ in the DIII-D tokamak. $\beta_N$ is given by an appropriately normalized ratio between the plasma pressure and the magnetic pressure and is a common figure of merit in fusion energy research. In addition to $\beta_n$ the state space contains its most recent change as well as the current power injection level and its most recent change. The action is the next change in the power injection level. The "ground-truth" dynamics for this problem are given by a neural network model learned from data processed as in Abbate et al. (2021). Control is done at a timestep of 200ms and the reward function is the negative absolute deviation from $\beta_n = 2$. Reliably controlling plasmas to sustain high performance is a major goal of research efforts for fusion energy, and though this is very much a simplification of the problem, we intend to extend and apply BARL to more realistic settings in the immediate future.

