# OpenReview forum: "An Experimental Design Perspective on Model-Based Reinforcement Learning"
_ICLR.cc/2022/Conference — ICLR 2022 Poster_

### Official Review · Reviewer_n6c6 · 2021-10-25

**Correctness:** 4
**Technical Novelty And Significance:** 3
**Empirical Novelty And Significance:** 3
**Recommendation:** 8
**Confidence:** 3

**Main Review:**

Strengths:
- Simple and easy to understand method for active learning in TQRL, that is based on the principle of expected information gain (EIG) from Bayesian experimental design. While not optimal, similar greedy maximization approaches have a strong track record in related problems such as active information gathering.
- A nice reformulation of the problem as reducing uncertainty about the optimal trajectory, which leads to a decomposition according to the sources of uncertainty.
- Empirically demonstrated to reach good performance with only a handful of samples, outperforming model-free methods and in some cases model-based competitors such as PETS.

Weaknesses:
1. The empirical evaluation only concerns simple domains where a Gaussian process (GP) model easily suffices. It is not clear if the methodology will scale to larger or more complex models, where evaluation of the EIG objective function itself might pose challenges.
2. To the best of my understanding, the performance of the proposed method is also capped to the performance of an MPC solution in a problem. This also shows in the results, e.g., Fig. 3 showing PETS on beta tracking and reacher obtaining a higher final reward, SAC & TD3 in other domains. While this is not necessarily an issue in the domains considered in the paper, generally MPC cannot be guaranteed to reach the same performance as an optimal solution. This is a limitation of the proposed approach.
3. There are a few points that need additional clarification, details below.


Detailed comments:


Questions on Algorithm 1:
- Why is # of points for optimization k an input when k does not appear in the algorithm?
- Line 2: does U refer to uniform distribution? In the initialization, should s_0 be sampled from p_0?
- Line 7: I don't understand why X is drawn from an uniform distribution over S x A. Then, X would be a pair of a state and an action. This does not really make sense for how it is used on Line 8 as a set.

Presentation of the results:
- Fig 1 (b), the colored trajectories are distracting, are they necessary?
- The significance of the colors in Fig. 2 is not clear.
- Could you clarify what exactly is the MPC method shown in Table 1, Fig. 3? I did not get it from the paragraph "Comparison Methods".



Other/Minor:
- This comment is just for information and needs no response: Considering the GP models, submodularity arguments might be helpful to further justify the greedy sampling approach. See, e.g., Krause et al. "Near-optimal sensor placements in Gaussian processes: Theory, efficient algorithms and empirical studies", JMLR 2008.
- Just before Eq. (1), there range 1, ..., H for the index i has a mismatch. That is, a_0 is not accounted for and no action should be drawn at step H.
- "The target the minimal information..." -- sentence seems ill-formed, please revise
- Eqns. (3)-(4): to remove any guessing, please mention here that blackboard H is the entropy
- "Given sufficient computational resources and an accurate model, model-predictive control will typically find very good if not optimal actions in an MDP." -- could you point to a source for this claim and/or make precise what this means? AFAIK, no error bound for the performance of MPC compared to an optimal solution is known.

**Added after the author response**
I thank the authors for their response, which clarifies my remaining concerns. I have increased the score to reflect this.

**Summary Of The Paper:**

This paper proposes an active learning method for transition query RL (TQRL) in continous domains. In TQRL, an agent queries the dynamics of the transition function at arbitrary state-action pairs in order to estimate the system dynamics. The objective is to query pairs that are most useful in solving the underlying control problem. The paper formulates TQRL as an expected information gain (EIG) maximization given the current dataset of observed state transitions, and applies posterior function sampling to evaluate EIG. The method is empirically evaluated in five continuous control tasks, with improvements in empirical sample complexity.

**Summary Of The Review:**

This paper proposes a simple but seemingly effective method for model-based RL in cases where obtaining transitions for training from the real environment is extremely costly. The method is based on well-founded principles of Bayesian experimental design. The paper itself is clearly written and easy to understand, and technically sound. The empirical results sufficiently clearly show the advantages of the proposed methods. More realistic experimental settings could be considered in future work.

---

> ### Author Response · Authors · 2021-11-15
> **Response to Reviewer n6c6**
>
> Thanks for your comments and feedback—we have incorporated these into an updated version of our paper (recently reuploaded). We appreciate the positive feedback and will respond to your concerns below.
>
> **Scaling:** We agree that in this work, we did not focus on high dimensional RL problems, primarily due to the choice of our GP model. Instead, our focus was on presenting the TQRL, \\(\text{EIG}_O\\), and BARL ideas as straightforwardly as possible using a model that allows us to easily compute the quantities involved. In principle, our procedure for computing the \\(\text{EIG}_O\\) acquisition function should be applicable to Neural GPs and possibly other probabilistic NN models without much modification. Though we do not pursue this in the present work, we intend to explore scaling BARL to these models in future work.
>
> **MPC as a limit on performance:**  Broadly, the goal of our work was to find a policy within a limited budget of queries that performs well on the MDP. Framed in this way, we are willing to trade an algorithm with high asymptotic performance for an algorithm that reaches a reasonable asymptotic performance in far fewer samples. In principle, any method of approximating the optimal policy on the posterior function samples could be used, including those that are more complicated but often reach higher performance. However, in this work, we chose to use MPC as it is the simplest control technique that would allow us to highlight the acquisition function idea.
>
> Additionally, we ran the following experiment to better understand the sensitivity of \\(\text{EIG}_O\\) to the approximation of the optimal policy: for a fixed problem and dynamics model, the MPC policy approximates the optimal policy in a way that roughly monotonically improves as the hyperparameters controlling the compute requirements increase. We run BARL on two of our example problems with a range of these hyperparameters and analyze its performance as the approximation to the optimal policy improves or degrades. At test time, we always use the best hyperparameters.
>
> We observe that although the policies used to generate \\(O\\) are clearly suboptimal, we see almost no degradation in the performance of the acquisition function, showing that \\(\text{EIG}_O\\) is fairly robust to the assumption \\(\pi_T \approx \pi^*\\) being violated. We have included these plots in appendix (Section B.1, Figure 5).
>
> **Notation in the algorithm box:** Regarding U, k, and X, we agree that the notation is imprecise. The actual procedure is to sample a set X of k points from the uniform distribution over S x A and choose the one that maximizes \\(\text{EIG}_O\\). We have updated our paper to make the presentation of the algorithm consistent with your points and correct in usage of mathematical types. Our initial sample is drawn uniformly at random from S x A.
>
> **Presentation:** We hoped that the colored trajectories in Figure 1b would demonstrate that there is a wide distribution over O that they are sampled from. We believe, in that light, they are worth including in the paper, and have added additional explanation. The colors in Figure 2 are simply to disambiguate different trajectories. We have added a comment in the figure caption which clarifies this.
>
> **Clarification on MPC:** Our MPC method collects points in the standard (rollout) RL setting by choosing actions via optimizing future rewards with transitions predicted by the mean function of the GP using equation 2. At every timestep the first action of the optimal sequence found is executed and the GP is conditioned on the new datapoint. At test time, we simply run model-predictive control on the mean of the GP, as in BARL.
>
> **Theoretical results:** Thanks for the suggestion to include submodularity arguments and in particular for the reference. We are interested in proving formal theoretical results for our method, and have been thinking along those lines. This suggestion definitely helps us move in the right direction.
>
> **Optimality of MPC:** Thanks for pointing out this statement about MPC. We agree that this statement is not as precise as we thought, so we have removed it from the updated draft.
>
> Finally, if you find our response satisfactory, we respectfully ask that you consider increasing your score. If it is still unsatisfactory, please let us know if there is anything else that we can do or clarify to improve this paper.

---

> > ### Comment · Reviewer_n6c6 · 2021-11-30
> > **Thank you**
> >
> > Dear authors, thank you for addressing my concerns in your response. This is just to let you know I have read it and adjusted my score accordingly.

---

### Official Review · Reviewer_AKFY · 2021-11-02

**Correctness:** 3
**Technical Novelty And Significance:** 3
**Empirical Novelty And Significance:** 3
**Recommendation:** 5
**Confidence:** 4

**Main Review:**

The main idea of the paper is really interesting. Most model based RL approaches are based on data efficiency, so clearly the idea of applying active learning is interesting. Furthermore, the paper focuses on active learning for policy optimality, instead of the most traditional exploration/exploitation approach.

However, I have trouble following the approach in the TQRL setting that the authors proposed. In that setting, there is already a generative model/oracle/simulator that allows querying arbitrary transitions. If we already have a transition oracle that we can query, what is the point of learning a transition model? Most planning approaches, like the ones presented in the paper and compared against are based on MPC with sampling of the transition model. If the control part was done in a fashion that required the full distribution, I could imagine some possible contribution, but given the use of CEM, it seems that the paper is trying to solve a problem that it is already solved at the beginning. For example, the 3 papers that the paper cites to justify the TQRL setting, one of them is for planning, not RL, and the other 2 are about theoretical analysis on sample complexity. There is not a practical implementation in an RL setting.

Being familiar with the RL and BO literature, the paper can be hard to follow. For example, the acquisition function is presented (s4) before the context where it is applied (s5). Also, in the Algorithm it is hard to find where the reward is optimized until the reader realizes that the 3rd line in the loop (executing pi) implies computing the optimal policy by MPC/CEM and then, generating the trajectories. It is not just executing a policy. The acquisition function also needs more clarification: the authors choose to compute O instead of a fully defined pi* to avoid sampling unnecessary states that are never visited from p0. This affirmation implies that both the initial distribution and the transition distributions are considerably limited. If the initial state is unknown or the noise level is reasonably high, the advantage of using O vanishes, because the trajectories end up visiting most of the state space. But the choice of O might have some limitations: what happens if the initial policy/trajectories are completely wrong, driving the queries to irrelevant state/actions for the goal? Is there any convergence guarantee?

The state of the art seems to miss an important part of literature on intrinsic motivation, curiosity, which have addressed similar problems before. I recommend the works of Pierre-Yves Oudeyer, Jürgen Schmidhuber, etc. For example, the work on Lopes et al. seems very related to the paper at hand.

Lopes M, Lang T, Toussaint M, Oudeyer PY. Exploration in model-based reinforcement learning by empirically estimating learning progress. InNeural Information Processing Systems (NIPS) 2012 Dec 3.

Furthermore, given that the surrogate model in this paper is based on GPs, there should be a mention and comparison to GP-based model based RL methods, such as PILCO, which has been, and they still are, the standard of sample efficient RL.

Deisenroth MP, Fox D, Rasmussen CE. Gaussian processes for data-efficient learning in robotics and control. IEEE transactions on pattern analysis and machine intelligence. 2013 Nov 4;37(2):408-23.

In fact, the MPC methods that the authors include in the comparison, seems to be a stripped down implementation of this work.

Kamthe S, Deisenroth M. Data-efficient reinforcement learning with probabilistic model predictive control. InInternational conference on artificial intelligence and statistics 2018 Mar 31 (pp. 1701-1710). PMLR.

In contrast, it seems that the inclusion of model-free methods in the comparison just adds noise to the plots. Maybe you can include the performance in the table and supplementary material and, instead, include GP-based RL in the plots, like those mentioned before.

Minor details:

-For fair comparison, given that the competitors cannot exploit the transition oracle, it would be interesting to see performance BARL in environmental steps, including the steps needed to reach the desired query. That is, the algorithm is able to sample arbitrarily any transition, but the agent has to reach it sequentially.

-The most complex problem in terms of exploration space seems to be the reacher which is 8D. How does the algorithm behave in higher dimensional problems, such as half-cheeta, ant, humanoid, etc. Or even some exploration challenging scenarios such as the ant-maze.

-The authors use EIGt as a proxy for the exploration algorithms. Why not use the full algorithms instead? For example, the code of Shyam et al is available.

-What is the implementation of the cartpole swing-up? Why haven't the authors used a standard benchmark such as the OpenAI gym version with mujoco or pybullet?

-Second paragraph, third line -> repeated “on”

Updated review:
----
After reading the updated version of the paper, I see the relevance of the proposed method and the TQRL setup: The paper is now much more readable and I have updated my score accordingly.

My only remaining major concern is with the comparison. Being a novel setup I understand the complexity of a fair comparison and the addition of PILCO clearly solves some of the questions. However, there are still some questionable choices. For example, in the submitted code, there are multiple versions of the cartpole and it is unclear which one is used, and the fact that there are some versions with the names of specific methods could be interpreted as different versions where used (which I assume was not the case). It's true that the OpenAI gym cartpole is not swing-up, but there is one in pybullet and there are many well known benchmarks such as dm_control. Using previously defined, unaltered tasks, like the Pendulum and Reacher would make all the experiments much more stronger.

**Summary Of The Paper:**

This paper presents an active learning algorithm for model based reinforcement learning. The algorithm is based on a new acquisition function that finds the most informative transition to query an oracle model in order to achieve better performance with respect to the optimal task.

**Summary Of The Review:**

The main idea of the paper is interesting and relevant for the field. However, I feel that the setting and the comparison needs to be revisited to have practical application.

---

> ### Author Response · Authors · 2021-11-15
> **Response to Reviewer AKFY (1/2)**
>
> Thank you for the review and comments on the paper—we have incorporated your feedback into an updated version of our paper (recently reuploaded). We’re glad you found the idea of active learning in RL interesting and we appreciate your detailed feedback. We’ll address your comments roughly in order.
>
> **TQRL setting motivation:** We believe the TQRL setting is well-motivated by real-world applications. Our study of the setting was originally motivated by an application of reinforcement learning that is currently being conducted by our colleagues in physics: they aim to find a controller that keeps plasma stable and improves performance in nuclear fusion reactors. They have access to a simulator which runs plasma dynamics 12,000x slower than real time (each transition takes on the order of 40 minutes). Running methods like CEM, which requires thousands of transitions, on this simulator would be infeasible. Thus, in this expensive setting, it is important to minimize the number of transition queries to find a good control.
>
> (Note that in our method, we solve this issue by running CEM on a cheap probabilistic model *only*, in order to determine where to effectively sample the expensive (real) transition and in order to choose actions at test time.)
>
> There are many applications with similar problem structures involving expensive transitions—in particular, the design and control of complex systems such as aircraft or logistics, the simulation of power systems scaling up to the electrical grid, chemical process engineering, and other expensive simulators in the physical sciences and robotics.
>
> **TQRL setting prior work:** We agree that there does not exist prior work addressing the TQRL setting in continuous MDPs. However, we see this as a novel contribution of our paper: we are the first to give an algorithm that is suitable for TQRL in the continuous case. It was not obvious to us when first presented with the real-world application mentioned above how best to take advantage of the simulator, and in our paper we formalize this problem and provide an initial solution.
>
> **Clarifying \\( \pi_{T’_\ell} \\):** In order to address your concern about reward optimization and “just executing a policy”, we have updated our paper to state more explicitly that executing \\(\pi_{T’_\ell}\\) is meant to use the notion of \\(\pi_T\\) defined in section 3.2 of a policy given by the solution to an optimization problem.
>
> **Choice of O in acquisition function:** We agree that if the occupancy distribution of the optimal policy was uniform over the space, using O would be no better than using \\( \pi^* \\) or T. We see in our experiments that \\(\text{EIG}_T\\), which samples according to the uncertainty of the model, thus treating all the points in the space as equally important, is not a good method of acquiring data for many of our problems. In the limiting case where the occupancy distribution of the optimal policy is uniform over the state space, our acquisition function would be identical to a strategy doing information gain over \\(\pi^*\\).
>
> **PILCO:** We agree that PILCO and related papers are relevant works. We have added them to our related work section. In fact, we have been working to add PILCO as a baseline for this study since well before the deadline but have had considerable difficulty in doing so. Currently, both implementations that we have tried suffer from numerical instabilities, and when they run smoothly often give poor performance on our benchmark problems. We are working to tune hyperparameters in order to achieve higher performance before adding these results to our paper. We plan to update the paper with these results once this is complete.
>
> **Related works:** Thanks for suggesting the works by Schmidhuber, Oudeyer, and the Lopes citation in particular. They are quite relevant and we’ve included them in our paper.
>
> **\\(\text{EIG}_T\\) baseline:** We claim that \\(\text{EIG}_T\\) is in fact equivalent to the Shyam et al. (2019) method with two modifications:
> 1. We use the same gaussian process dynamics model that we use for BARL and MPC in our evaluation.
> 2. We allow \\(\text{EIG}_T\\) to operate in the TQRL setting.
>
> We intentionally included these modifications as we believe them to provide a better comparison to BARL as well as being likely to improve the performance of the \\(\text{EIG}_T\\) method itself—this is due to a more general access to the transition function and a model that seems to perform better on these lower-dimensional problems.
>
> **Scaling:** We agree that in this work, we did not focus on high dimensional and complicated RL problems, as our main goal was to present TQRL, \\(\text{EIG}_O\\), and BARL ideas as straightforwardly as possible. In future work, we intend to extend BARL/\\(\text{EIG}_O\\) to more scalable models, such as neural GPs, ensembles of neural networks, or BNNs, in order to address more complicated tasks.

---

> > ### Author Response · Authors · 2021-11-15
> > **Response to Reviewer AKFY (2/2)**
> >
> > **Cartpole:** There is not a cartpole *swing-up* problem included in the standard gym package as far as we know (the gym cartpole is a cartpole *balance* task, involving simply trying to balance an already mostly upright pole). A full implementation of the cartpole swing-up environment that we used is included in our code submission.
> >
> > **Writing:** Thanks for the typo suggestions, we have fixed as many as we have found in our updated paper!

---

> > > ### Author Response · Authors · 2021-11-23
> > > **Updated Paper with PILCO Results**
> > >
> > > **PILCO:**
> > > We were able to get the PILCO implementation working on our Pendulum task, and have added these results to our updated paper in Section 6, Figure 3 (and we are in the process of adding results for the other tasks). Here, we find that PILCO reaches good performance within a small number of trials, though the sample efficiency and asymptotic performance does not exceed that of MPC and BARL.
> > >
> > > With the addition of PILCO, and the points above, we hope that we have addressed the major concerns in your review.

---

### Official Review · Reviewer_X5up · 2021-11-04

**Correctness:** 3
**Technical Novelty And Significance:** 3
**Empirical Novelty And Significance:** 3
**Recommendation:** 8
**Confidence:** 4

**Main Review:**

Strengths:

-The paper provides a principled and interesting approach to an impactful problem. The idea of approximating the optimal policy and then computing information gain on the sequence of states visited by the optimal policy is both novel and clever.

Weaknesses:

-The paper uses independent GPs to model the transition dynamics of the MDP. Could multi-task GPs be used here to improve the model’s capability? For example, [1] allows for  high-dimensional, correlated outputs, which could be used to scale your method to larger problems.

-I’m curious about the sensitivity to the approximation of the optimal policy. For example, such a study could be carried out on a domain where the optimal policy is computable and then various approximations to it (e.g., MPC, MCTS, myopic policies) could be used as the approximation.

-Even though the motivation is for problems with expensive transitions, wall times for the new method should be included so that readers can get a sense for how computationally expensive the method is.

-There are references that might be relevant. Please see [2, 3] from the OR literature.

[1] https://arxiv.org/abs/2106.12997
[2] https://pubsonline.informs.org/doi/abs/10.1287/opre.2018.1772
[3] https://pubsonline.informs.org/doi/abs/10.1287/opre.1100.0873


**Summary Of The Paper:**

This paper considers the very relevant (in my opinion) problem of data-efficient RL and is potentially applicable in nearly all situations where RL is to be applied in the real-world rather than a simulator. The authors propose an acquisition function based on expected information gain (EIG) approach. The overall framework consists of a (1) Gaussian process model of the transition dynamics, (2) an MPC approximation of the optimal policy, (3) the EIG acquisition function where the target variable is the sequence of states visited by the optimal policy. Empirical results are shown for a number of continuous control domains.

**Summary Of The Review:**

The paper develops a novel method for an important problem. I, however, have a few comments above that I believe would improve the paper.

---

> ### Author Response · Authors · 2021-11-15
> **Response to Reviewer X5up**
>
> Thank you for the feedback! We have incorporated this into an updated version of our paper (recently reuploaded). Thanks also for the positive feedback. We’ll respond to the negative comments here and have updated the paper accordingly.
>
> **Modeling Suggestions:** We agree that the model could be improved, either by using multi-task GPs or by using probabilistic neural network methods. However, in this work, we aimed to present the TQRL, \\(\text{EIG}_O\\), and BARL ideas as straightforwardly as possible using a model that allows us to easily compute the quantities involved. We are currently exploring the use of other models as future work.
>
> **Accuracy of the optimal policy estimate:** Thanks for your question about sensitivity to the approximation of the optimal policy. To answer this, we have performed an additional experiment as follows: for a fixed problem and dynamics model, the MPC policy approximates the optimal policy in a way that roughly monotonically improves as the hyperparameters controlling the compute requirements increase. We run BARL on two of our example problems with a range of these hyperparameters and analyze its performance as the approximation to the optimal policy improves or degrades. At test time, we always use the best hyperparameters.
>
> We observe that although the policies used to generate \\(O\\) are clearly suboptimal, we see almost no degradation in the performance of the acquisition function, showing that \\(\text{EIG}_O\\) is fairly robust to the assumption \\(\pi_T \approx \pi^*\\) being violated. We have included these plots in appendix (Section B.1, Figure 5).
>
> **Computational cost:** We have also included a table with the wall time per iteration in the appendix in the updated version of our paper (Section A.1, Table 3), along with some discussion of how much each step of the process costs.
>
> **References:** Thank you for the additional references, we agree that they are relevant and have included them in the updated version of our paper.
>
> Finally, if you find our response satisfactory, we respectfully ask that you consider increasing your score. If it is still unsatisfactory, please let us know if there is anything else that we can do or clarify to improve this paper.

---

> > ### Comment · Reviewer_X5up · 2021-11-29
> > **Response**
> >
> > Thank you for addressing my comments! The results about the approximation to the optimal policy was nice to see. I also am curious to see more about the approximation (now I am wondering: what if the policy is very bad, would we still get reasonable results due to some other reason?) I will update my review.

---

### Official Review · Reviewer_gjNV · 2021-11-06

**Correctness:** 4
**Technical Novelty And Significance:** 3
**Empirical Novelty And Significance:** 2
**Recommendation:** 8
**Confidence:** 3

**Main Review:**

## Strengths

1. The paper is very well written.

2. The proposed framework and algorithm are well motivated with repeated grounding in practical considerations around problems where a ground truth model exists but is expensive to compute samples from. Independent of empirical evaluation, there are reasonable intuitive arguments for why one might expect improvements in sample complexity.

3. A wide variety of relevant baselines were compared against, including some ablations of the proposed algorithm. The empirical results presented appear promising.

## Weaknesses

1. The current setup they used for BARL appears relatively computationally expensive (e.g., using GPs to model the dynamics, running MPC over every posterior transition function, etc.)

2. The domains are on the simpler end, which are reasonable for the scope of the paper but leave questions about how the approach might scale to larger domains.

3. Only five random seeds were evaluated. Recent criticisms have emphasized how five seeds might not be representative of the data distribution- can the authors comment on the statistical significance of the results? Along those lines, I couldn't find in the text what the shaded regions represent in Figure 3? Can the authors further comment on how hyperparameters were selected across each algorithm and whether they provide a fair comparison?

## Minor stuff which didn't affect the review:

"the the" near the start of Section 3.1
"evaluatation episdoes" near the start of Section 6

**Summary Of The Paper:**

This paper uses insight from Bayesian optimal experimental design and define an *acquisition function* which quantifies information gain about an MDP's optimal solution given a state-action pair. Such an acquisition function would provide a mechanism for selecting which transitions to query from a ground truth model in model-based RL, in hopes to greatly improve sample efficiency of a learning algorithm. They propose a method for computing this acquisition function, and evaluate it empirically.

**Summary Of The Review:**

Taking the above into account, I recommend acceptance of the paper. While I feel the empirical evaluation could be improved, I find the motivation and intuitive arguments behind the proposed algorithm convincing enough to be a useful contribution in the literature.

---

> ### Author Response · Authors · 2021-11-15
> **Response to Reviewer gjNV**
>
> Thank you for your review and comments. We have incorporated them into an updated version of the paper (recently reuploaded). We appreciate your positive feedback and will address your concerns roughly in order.
>
> **Computational cost:** We agree that BARL in its current form is computationally expensive. Our main goal in this work is to address the problem of sample efficiency in RL, even at the cost of using additional compute. This is appropriate in problems where the cost of a transition query is large enough that the compute we spend on the acquisition function is negligible. That being said, we agree that more information about compute costs would be helpful to readers. We have included a table in the appendix (Section A.1, Table 3) detailing the time per iteration of BARL on each of our tasks.
>
> **Complexity of experimental tasks:** We also agree that the domains we evaluate BARL and competitors on in this work are not as complicated as certain high-dimensional RL applications. In this work, we aim to lay out the basic ideas of TQRL, \\(\text{EIG}_O\\), BARL, and acquisition functions for reinforcement learning—we will certainly explore how far we can push these techniques in the future. The largest barrier to scaling up to more complicated problems is the GP. Our method is in principle applicable to any kind of probabilistic model. We believe that the path to scaling is to swap the GP for more scalable models (including well-calibrated probabilistic neural models), which we are currently exploring. However, we leave this further exploration of modeling choices to future work.
>
> **Experimental Setup:** The shaded regions of our learning curves represent the standard error of the mean performance across our random seeds. We believe the curves and their associated error regions are well-separated, implying a robust effect from applying BARL. Per your suggestion, we are re-running some of our experiments with 20 seeds and will update the paper when those experiments complete. As we discussed in the appendix, we manually chose hyperparameters for our algorithm which reached a reasonable tradeoff between computational performance and MDP performance. For competitor algorithms we used the recommended hyperparameters of the implementation unless the results didn’t look reasonable, at which point we performed light hyperparameter tuning. In fact, PETS uses much larger planning budgets than BARL.
>
> **Writing:** Thanks for pointing out the typos—we have fixed each of them in our new draft.

---

### Author Response · Authors · 2021-11-23
**We have updated our paper.**

We thank the reviewers for their valuable feedback on this paper. We appreciate that the reviewers thought our paper was “very well written” and “well motivated with repeated grounding in practical considerations” (gjNV); that the idea “is both novel and clever” and “a principled and interesting approach to an impactful problem” (x5up); and a “simple method which is also shown to be quite effective” that shows “good performance with only a handful of samples” (n6c6)​​.

We worked to address each of the reviewers’ questions. In particular, we have added three additional experiments to our paper:
- Reviewers had questions about the optimal policy for posterior function samples found by MPC. To answer this, we conducted an additional experiment where we varied the compute budget available to the controller and found our algorithm robust to a poor approximation of the optimal policy.
- Another common question was about the practical runtime of our method. We have added a table to the appendix detailing how much time is spent on each step of our method across each of our benchmark problems.
- One reviewer also mentioned that PILCO would be a good benchmark to include. We have added PILCO results for one of our tasks and are in the process of getting results for the other tasks as well.

One reviewer had concerns about the motivation for the TQRL setting. In our response, we described a few concrete applications where TQRL is applicable and valuable, including real use cases in the physical sciences which directly motivated this work. In particular, our method is currently being applied by our collaborators in this setting using an expensive physical simulator of plasma dynamics.

---

### Decision · Program_Chairs · 2022-01-20

**Decision:**

Accept (Poster)

**Comment:**

In this paper, the authors introduce an exploration method for RL according to experimental design perspective via designing an acquisition function, which quantifies how much information a state-action pair would provide about the optimal solution to a Markov decision process, and the state-action that maximizes such acquisition function will be used for sampling for policy update. The empirical evidences show the proposed method is promising.

Since most of the reviewers support the paper, I recommend acceptance of this submission.

However, besides the questions raised by the reviewers, e.g., computation cost and planning quality from CEM, there is a major issue need to be clarified in the paper:

>The algorithm designed for RL with generative model, which makes the state-action reset can be conducted (this is sometimes impossible in practice where the agent must start from initial state). This is different from the common RL setting, and thus reduce the complexity of RL. This should be emphasized in the paper. Meanwhile, for a fair comparison, this should be explicitly specified in experiment setting.